# Combined Training with Aerobic Exercise Performed Outdoors Can Promote Better Blood Pressure and Affective Responses in Individuals with Cardiovascular Risk Factors

**DOI:** 10.3390/ijerph192316009

**Published:** 2022-11-30

**Authors:** Janara Antunes De Moraes, Guilherme Tadeu De Barcelos, Juliana Cavestré Coneglian, Bárbara Carlin de Ramos Do Espírito Santo, Rodrigo Sudatti Delevatti, Aline Mendes Gerage

**Affiliations:** Post-Graduate Program of Physical Education—UFSC, Federal University of Santa Catarina, Florianopolis 88040-900, Brazil

**Keywords:** combined training, aerobic exercise, blood pressure, affective responses, cardiovascular risk

## Abstract

The purpose of the study was to compare the effects of two models of combined training (CT) (aerobic and resistance exercise realized in the same training session), with aerobic training performed in different environments (indoor or outdoor), on blood pressure (BP), heart rate (HR), and affective response in individuals with cardiovascular risk factors. Twenty-six participants were allocated, in a non-randomized design, into CT with aerobic exercise performed indoors (ICT) or outdoors (OCT). Both groups were submitted to three weekly CT sessions, with aerobic exercises performed on ergometers or an athletics track. Before and after nine weeks of training, BP and HR at rest were measured. In the last session of the training, the affective response was collected. The individuals were 65.8 ± 7.8 (ICT) and 67.3 ± 8.2 (OCT) years. Lower values of diastolic BP were observed for the OCT group at post-training (*p* < 0.001). Moreover, in OCT, a significant inverse correlation was identified between the affective response to training and changes in systolic BP (r = −0.60; *p* = 0.03) and mean BP (r = −0.62; *p* = 0.02). In conclusion, CT, with aerobic exercise performed outdoors, seems to be more effective in reducing BP with better affective responses to training.

## 1. Introduction

The growth of the elderly population is a worldwide phenomenon. Today, there are 962 million people aged 60 and over, which is equivalent to 13% of the world’s population [1]. According to Chen et al. (2020) [2], aging is a complex natural phenomenon, characterized by the decline in structure and function, and a decreased adaptability and resilience, ending in death. According to Schmeer et al. (2019) [3], one of the main negative aspects inherent to the aging process is the increased susceptibility to develop chronic noncommunicable diseases (NCDs), highlighting, in this scenario, hypertension, which affects, worldwide, 32% of women and 34% of men aged between 30–79 years [4], and more than 60% of people over 60 years old [5].

Hypertension is considered a multifactorial disease [6] and, among its strategies of prevention and treatment, it is emphasized the practice of physical exercises. Different training modalities have been shown to be effective in reducing BP, such as aerobic, resistance, and combined training (CT), which refers to the combination of aerobic and resistance in the same training model [7,8]. A systematic review with meta-analysis found reductions of 3.5 and 1.8 mmHg after aerobic and dynamic resistance training, respectively, and of 2.5, 3.2, and 2.2 mmHg after aerobic, dynamic resistance training, and CT, respectively [7]. Some studies have sought to explore the impact of manipulating different training variables, such as volume, intensity, training load, dose-response curve, and affectivity with the practice, in reducing BP, especially in the elderly [9,10,11,12,13,14].

In addition, the environment in which the exercise is performed (indoor or outdoor) has also been the subject of research in order to observe the influence of the environment on BP responses. Acutely, the study by Calogiuri et al. (2015) [15] with CT identified that, compared to the indoor group, the outdoor group showed a reduction in diastolic BP values. Chronically, Lacharité-Lemieux and Dionne (2016) [16] found that CT performed outdoors was more effective in lowering BP in postmenopausal women. In the study by Krinski et al. (2017) [17], BP values showed no difference between the outdoor and indoor training groups. However, these findings are not universal and should be more investigated, especially in clinic populations, which include patients with a cardiovascular risk profile.

Moreover, no study has investigated whether the internal load of training, measured by the Rating of Perceived Exertion (RPE) referring to the session (RPE-session) and the affective response to exercise interfere in the reduction of BP levels in different practice environments. By now, it is known that the RPE tends to be lower in exercises practiced outdoors [15,16,18] and that better affective responses have been observed when outdoor training is considered, possibly due to the fact that it promotes a greater sense of post-exercise tranquility and intention to continue practicing [17,18,19]. However, more studies relating these parameters of internal load and affectivity with possible physiological adaptations to training are needed.

Once the environment (indoor and outdoor) has been one of the variables that could be considered in the exercise prescription [16,20,21,22], and that positive affective response may be related to prolonged adherence to physical exercise [16], knowing whether aerobic training (AT) performed indoors or outdoors promotes different adaptations in BP and whether this is associated with the internal load and affectivity in response to exercise may help in better structuring of the training program, especially when the objective is the prevention or treatment of hypertension. From this perspective, the present study aimed to analyze the effect of CT, with aerobic exercise performed indoors and outdoors, followed by resistance exercises realized indoors, on the BP and heart rate (HR) of adults and elderly individuals with cardiovascular risk factors. In addition, we sought to analyze whether there is a relationship between the internal load of training and the affective response to exercise and the changes in BP and HR resulting from CT, with aerobic exercise performed indoors and outdoors. Our hypothesis is that CT is capable of reducing BP in adults and elderly people with cardiovascular risk factors, regardless of the place where the aerobic exercise is performed, but outdoor practice can potentiate these effects, which should be associated with affective responses provoked by exercise.

## 2. Materials and Methods

### 2.1. Design

The study is characterized as a non-randomized comparative clinical trial that is part of a larger project, approved by the Ethics and Research Committee involving humans at the Federal University of Santa Catarina (UFSC) (opinion number: 3,615,659) and registered in the ReBEC platform (registration: RBR-6sz5xr).

### 2.2. Recruitment and Participants

Participants were adults and elderly (≥60 years old) of both sexes, members of the Cardiorespiratory Prevention and Rehabilitation Program (PROCOR), a training program with a pragmatic approach. As eligibility criteria, the participants should have been participating in PROCOR for at least three months and could not have osteomioarticular limitations that could hinder the execution of the exercises. They should have cardiovascular risk factors, which included the presence of diabetes, dyslipidemia, or hypertension, either singly or in combination. The recruitment of the participants was performed in a non-randomized design, after the disclosure of the study in PROCOR. All individuals who met the inclusion criteria were invited to participate in this study and those who accepted the invitation, after being informed about the procedures, risks, and benefits of the study signed the Informed Consent Term.

### 2.3. Experimental Procedures

Initially, the participants answered a medical history, in which sociodemographic information was collected to characterize them. After that, they were submitted to three non-consecutive days of resting BP collection (pre-training) and, after that, they were divided into two groups (OCT and ICT), both submitted to nine weeks of CT. During the training program, two affective responses and one internal load (RPE) collection took place in the last session and in the first and last week of the training program, respectively. At the end of the nine-week training program, the participants’ resting BP was evaluated again on three non-consecutive days (post-training).

### 2.4. Allocation

The allocation of participants to the OCT and ICT groups was performed intentionally, considering the time available for practice, due to weather, and lighting conditions of the outdoor environment. Thus, it was necessary to allocate the training sessions that usually performed their activities from 5 pm to 6 pm to OCT, and the sessions that performed their activities from 6 pm to 7 pm to ICT.

### 2.5. Evaluations

#### 2.5.1. Sample Characterization

Sociodemographic data (sex and age) and health status (presence of diseases and use of medication) were collected from the anamnesis. In addition, body mass and height were measured using a Marte^®^ scale and an AlturaExata^®^ stadiometer, respectively. From these data, the body mass index (BMI) of each participant was calculated.

#### 2.5.2. Blood Pressure Measurement

Resting BP and HR were collected pre and post-intervention, using automatic monitoring equipment (OMRON, model HEM-7113, Sao Paulo, Brazil), and according to instructions from the Brazilian Guidelines on Hypertension (2016) [23]. The collections were performed on three non-consecutive days (Monday, Wednesday, and Friday), before the training sessions. Before the measurement, all the participants of both groups remained seated at rest for 10 min in the same place (indoors). After this resting period, three BP and HR measurements were taken, with a one-minute interval between each measurement. For the analyses, the average of all the measurements taken each day was adopted.

#### 2.5.3. Training Internal Load Measurement

The RPE was applied to assess the internal load of training, using the Borg scale adapted by Foster et al. (2001) [24], in the first and last weeks of training. At the end of the CT session, individuals were asked “How intense was the training session today?”. The answer was given based on a scale from 0 to 10, considering that the higher the number chosen, the more intense the exercise session was. The participants were familiarized with the scale.

#### 2.5.4. Affective Response to Training Measurement

Hardy and Rejeski’s (1989) [25] feeling scale was used to determine the affective response to training. Each participant answered the question “How enjoyable was it to perform this exercise session?” at the end of the last session of the training program. The answer was given considering the scale ranging from +5 (“very good”) to −5 (“very bad”). Familiarization with this scale was done during the intervention, starting in the second week of training.

#### 2.5.5. Intervention

The intervention lasted nine weeks, during which the participants performed CT with indoor or outdoor aerobic exercise. The participants of the two groups had already been receiving the same training for at least three months in the PROCOR extension project, however, what differed was the practice of AT, in which one group performed indoors (ICT) and the other outdoors (OCT). Both groups received the same CT session, in which the exercise sessions throughout the nine weeks of training lasted 60 min and were performed three times a week on alternate days, distributed as follows: Session A (Monday), Session B (Wednesday), and Session C (Friday). Before the beginning of each session, each participant’s BP was evaluated as a safety measure, and then AT was performed, lasting 24 min, followed by resistance training (RT) lasting approximately 20 to 25 min. Stretching exercises were performed in the last 10 min of the session. The nine weeks of intervention corresponded to a macrocycle, divided into three mesocycles with three weeks each.

The training intensity throughout the macrocycle, for both modalities (AT + RT), was distributed as follows: in the first mesocycle, sessions A and C were performed at moderate intensity and session B at high intensity. In the second mesocycle, only session B was performed at moderate intensity, while sessions A and C were performed at high intensity, and in the last mesocycle, all sessions were performed at high intensity. The intensity modulation is represented below in Figure 1.

The AT was performed using the interval method on treadmills (ICT) or on the running track (OCT), applying in both groups, the RPE method with the Borg scale of 6 to 20 (BORG, 1982) [26] to control the intensity. In the case of unfavorable weather conditions, AT sessions took place indoors also for the OCT group. However, to minimize the interference of this approach in our study, this was performed minimally, while in extreme conditions. Both groups were told to change the intensity according to the time determined for the stimuli. The sessions were modulated as follows: (a) moderate: three minutes at intensity 11 (active recovery) and three minutes at intensity 15 (effort) on the Borg scale, until completing 24 min, with an effort/recovery ratio of 1:1; and (b) high: the stimuli alternated between two minutes at stage 11 (active recovery) and four minutes at stage 15 (effort), until completing 24 min, with an effort/ recovery ratio of 2:1.

The RT was composed of exercises for the main muscle groups, executed dynamically, involving the body weight or alternative materials (elastic band and free weight). The exercises were distributed in the three weekly sessions as follows-session A: Upper limb exercises (incline push-up on the bar, standing unilateral row with a rubber band, shoulder press with a dumbbell, and reverse sit-up), session B: Upper and lower limb exercises (body weight squat, incline push-up on bar, body weight lunge, standing row with rubber band, and crunch), and session C: Lower limb exercises (body weight squat, body weight lunge, hip thrust, and crunch). In the moderate-intensity sessions, two sets of each exercise were performed, each set lasting 30 s with one minute in between. In the high-intensity sessions, three sets of each exercise were performed, each set lasting 20 s and one minute apart. All sets were performed in the maximal velocity of execution.

### 2.6. Data Analysis

Data were analyzed using the statistical package R, version 3.5.3 (Lucent Technologies, Orlando, FL, USA). The general characteristics of the study participants were presented by means ($\bar{x}$) and standard deviation (sd) for continuous variables, or absolute (*n*) and relative (%) frequency for categorical variables. Data normality was verified using the Shapiro-Wilk test. The two-way ANOVA for repeated measures was used for the comparison of groups (indoor and outdoor) and time (pre and post) regarding the changes in hemodynamic variables, after confirming the appropriate assumptions. Tukey’s post hoc test was applied to identify intra- and intergroup differences, when relevant. To assess the relationship between internal load and affective response with the hemodynamic variables of the training groups, Spearman’s linear correlation was applied. To calculate the effect size (ES) of the pre- and post- intervention moments, Cohen’s test was used, considering the mean and standard deviation values (COHEN, 1998) [27]. A significance level of 5% was adopted in all analyses.

## 3. Results

Thirty-one participants started the training program, but five participants were excluded from the final analysis; four participants were absent in the final collections for personal reasons and one participant was absent due to a stroke not associated with the intervention. Thus, 26 participants completed the research, and most of the sample was composed of elderly (84.6%) males (61.5%) with a mean BMI of 28.5 kg/m² and with hypertension (65.4%), with no difference between the groups regarding these general characteristics (*p* > 0.05). Adherence to the training sessions in the OCT group was 72.7% (±11.0), and in the ICT group was 65.7% (±13.0), with no significant difference between the groups (*p* = 0.199). Both groups tolerated the training protocols well, with no adverse events. There was no difference between the groups for the sample characterization variables at baseline (*p* > 0.05) (Table 1).

Table 2 presents the SBP, DBP, mean BP (MBP), and HR data of the OCT and ICT groups, before and after the nine weeks of training. An isolated effect of the group was identified for the variables DBP (*p* = 0.004) and MBP (*p* = 0.015), with lower values observed for the OCT group. In addition, a group vs. time interaction was identified for DBP (*p* = 0.040), with a statistically significant difference between groups at the post-training, with lower values observed for the OCT group (*p* = 0.0007). No intra or intergroup differences were identified for SBP and HR (*p* > 0.05). When analyzing the same variables considering only hypertensive participants, an isolated group effect was identified for DBP (*p* = 0.004) and MBP (*p* = 0.003), with no statistically significant difference for SBP and HR (*p* > 0.05).

Regarding the results of internal load, represented by the RPE in the pre- and post-training moments, no statistically significant difference (*p* > 0.05) was identified within or between groups (Table 3). When correlating the RPE with the deltas (∆) of SBP, DBP, MBP, and HR of the OCT and ICT groups, no statistically significant results were found in either group (*p* > 0.05).

Statistically significant differences were observed comparing the affective response to the training of the OCT and ICT groups, in which the OCT group showed a greater affective response compared to the ICT group (OCT = 4; ICT = 2; *p* = 0.003). Figure 2A–D presents the correlations between the deltas (∆) of SBP, DBP, MBP, and HR with the affective response to training in the OCT and ICT groups. An inverse correlation was observed for ∆ SBP (r = −0.60; *p* = 0.03) and ∆ MBP (r = −0.62; *p* = 0.02) in the OCT group, suggesting that participants who reported higher rates of affectivity to training showed greater reductions in SBP and MBP.

## 4. Discussion

The main findings of the present study indicated that: (a) CT with aerobic exercise performed outdoors seems to be more effective in reducing BP, especially DBP, in adults and elderly with cardiovascular risk factors who were already trained; (b) better affective responses are identified in CT sessions when aerobic exercise is performed outdoors and such affective responses are associated with greater reductions in BP in trained adults and elderly with cardiovascular risk factors. Our hypothesis was partially confirmed, considering these findings, which should be interpreted with caution, due to the limitations of the study, which will be presented below.

Several studies show that regardless of the environment that the aerobic exercise is practiced, CT is effective in reducing BP values in adults and the elderly [7,28,29,30,31]. It is noteworthy that most of these studies included untrained participants. In trained people with lower basal BP values (~123 mmHg), as the participants of our study, it may be more difficult to find changes in BP resulting from physical training, which explained, at least in part, our results regarding the absence of BP changes in the ICT group. In this sense, our findings indicate that the environment can influence this process, since the DBP of the participants in the OCT group was lower than in the ICT group after the intervention, suggesting that the CT with aerobic exercise performed outdoors seems to be more effective at reducing this variable. This result suggests that exercising in the same training structure, even with the same intensity (prescribed and confirmed by the measured internal load) causes different BP adaptations in trained people only due to the environmental factor. Similarly, in the study by Lacharité-Lemieux and Dionne (2016) [16], a significant reduction of 5.6 mmHg for DBP was observed in the OCT group, and no significant reduction in the ICT for this variable.

An important point to be observed in exercises performed outdoors are the benefits of sun exposure to the body. There is epidemiological evidence showing an association between vitamin D deficit and the development of hypertension [32]. On the other hand, the association between BP reduction and skin exposure to ultraviolet B radiation, considered the main source of Vitamin D, has also been reported in the literature [33]. However, further studies need to be developed considering the influence of the environment on this variable, and the role of sun exposure and Vitamin D in this process.

The importance of reducing DBP values in adult and elderly individuals is highlighted, especially in hypertensive individuals, considering that, as well as elevated SBP values, increased DBP contributes significantly to cardiovascular risk [34]. In addition, DBP is more associated with the resistance that the heart has to overcome to eject blood from the heart. Based on this assumption, a reduction in DBP could be linked to a reduction in vascular resistance, which is relevant, especially in the elderly who, as a result of aging, have higher arterial stiffness, which culminates in higher peripheral vascular resistance.

Moreover, in the present study, participants who reported greater affective response to exercise in the OCT group showed a greater reduction in SBP and MBP. Thus, it is suggested that performing the exercise in an external environment provides greater satisfaction, and a better sensation of pleasure, and this can attenuate vascular stress at the same physiological load. Corroborating these results, Lacharité-Lemieux; Brunelle; Dionne (2015) [35] identified that, after 12 weeks of training, the exercise-induced changes in affective valence were greater for the OCT group, and the feeling of post-exercise tranquility, possibly associated with the better affective response to exercise, increased for the OCT and decreased in the ICT group. Complementarily, in the aforementioned study, regarding adherence to training, the results were significantly higher for the outdoor training group, differing from the findings of the present study in which there was no difference between groups for this variable.

Affectivity has been shown as an important element to enhance the benefits of physical training, both in physical and emotional aspects [15,35,36]. It is suggested that contact with nature or outdoor environments positively influences the affective response to training, whether aerobic, resistance, or a combination of the two modalities [15,17,35,36,37]. A study conducted by Calogiuri, Nordtug, and Weydahl (2015) [15] demonstrated that individuals who practiced OCT, in addition to having higher affectivity to exercise, demonstrated higher intention to continue practicing an exercise compared to the ICT group. Although the intentionality of continuing to practice was not assessed in the present study, this is interesting data from the point of view of adherence to physical training.

Additionally, in the research conducted by Calogiuri, Nordtug, and Weydahl (2015) [15], the authors observed that the group that trained outdoors reported a lower RPE when compared to ICT. Other studies show that when exercising outdoors, individuals tend to present a lower RPE, since they can focus on the environment around them, taking their attention away from body stimuli [18,19], differing from the result found in the present study, in which the RPE after the training period showed no difference between the training groups. However, it is worth noting that, in terms of effectivity, in the present study, the CT, with the exercise performed in an external environment (outdoor), provided better responses, suggesting that, perhaps, this environment favors the feeling of well-being and pleasure. These findings are relevant from a practical point of view, since this greater affective response may favor the intentionality to continue with the practice. Moreover, our findings add to the literature the fact that higher levels of affectivity to OCT correlated with greater chronic reductions in BP, indicating an important relationship between affective responses and physiological adaptations to training.

The results of the present study have relevant practical implications that can improve the prescription of exercises for the public. It is suggested, for example, that the prescription to this public should be seen in a broader way in order to better explore outdoor spaces for the practice, such as squares, sidewalks, and places in the city suitable for running and walking. Furthermore, the exercise prescription should focus on providing benefits in emotional aspects as well, since such aspects, besides being able to favor greater adherence to training, have shown to be associated with the physiological and hemodynamic adaptations expected from the practice of training programs.

The present study has as strong points the infrastructure used for the execution of the training sessions, being used only a large room and a walking track, which could easily be applicable for the general public, since the walking track could easily be replaced by a walk in the park, square, or sidewalk, etc. Still, the materials used for the practice of CT also favor the applicability of the results, since some resistance exercises were performed with body weight or with the help of simple implements, such as rubber bands and dumbbells that can be easily acquired or adapted for everyone to have access.

Despite this, the study has some limitations that cannot be disregarded, mainly the sample size and the fact that the sample allocation was not randomized. Nevertheless, we calculated the sample power (a posteriori) and we achieved 0.96, 0.99, 0.99, and 0.98 for SBP, DBP, MBP, and HR. Moreover, the study sample consisted of normotensive and hypertensive participants, taking different medications. Therefore, we suggest future research with a larger and more homogeneous sample, preferably only with hypertensive patients. However, the analyses, even with a reduced sample size, involving only hypertensive patients resulted in the same findings when compared to those identified when the entire sample was considered. In addition, the lack of control over food intake can also be considered a limitation of the study, considering the influence of this factor on BP levels.

## 5. Conclusions

Based on the results, it is concluded that nine weeks of CT, with aerobic exercise performed outdoors, seems to be more effective in the reduction of DBP in adults and elderly with cardiovascular risk factors if compared to the CT, with aerobic exercise performed indoors. Furthermore, outdoor aerobic exercise seems to promote higher levels of effectivity, with an identified correlation between greater affective response and greater reduction in SBP and MBP levels. Thus, it is suggested that the aerobic exercise practice environment should be considered in the development of a prescription for CT for this public since the outdoor practice environment seems to promote better affective responses and better chronic reductions in BP. However, these results of the present study should be carefully interpreted since there was no within-group effect of training for BP values.

## Figures and Tables

**Figure 1 ijerph-19-16009-f001:**
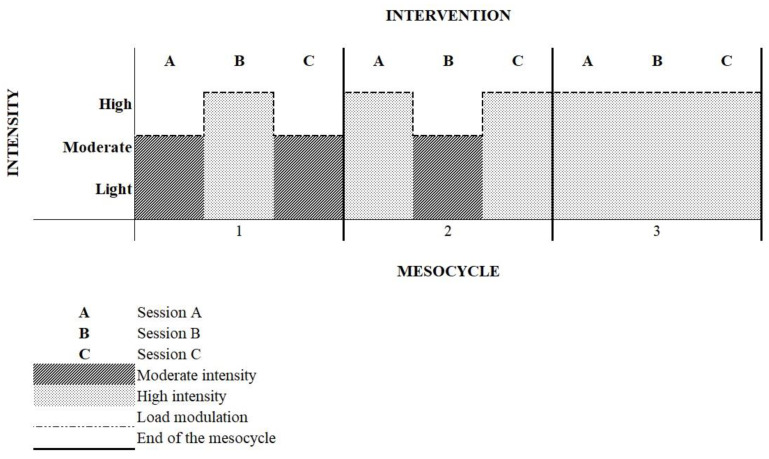
Intensity modulation during the nine weeks of training.

**Figure 2 ijerph-19-16009-f002:**
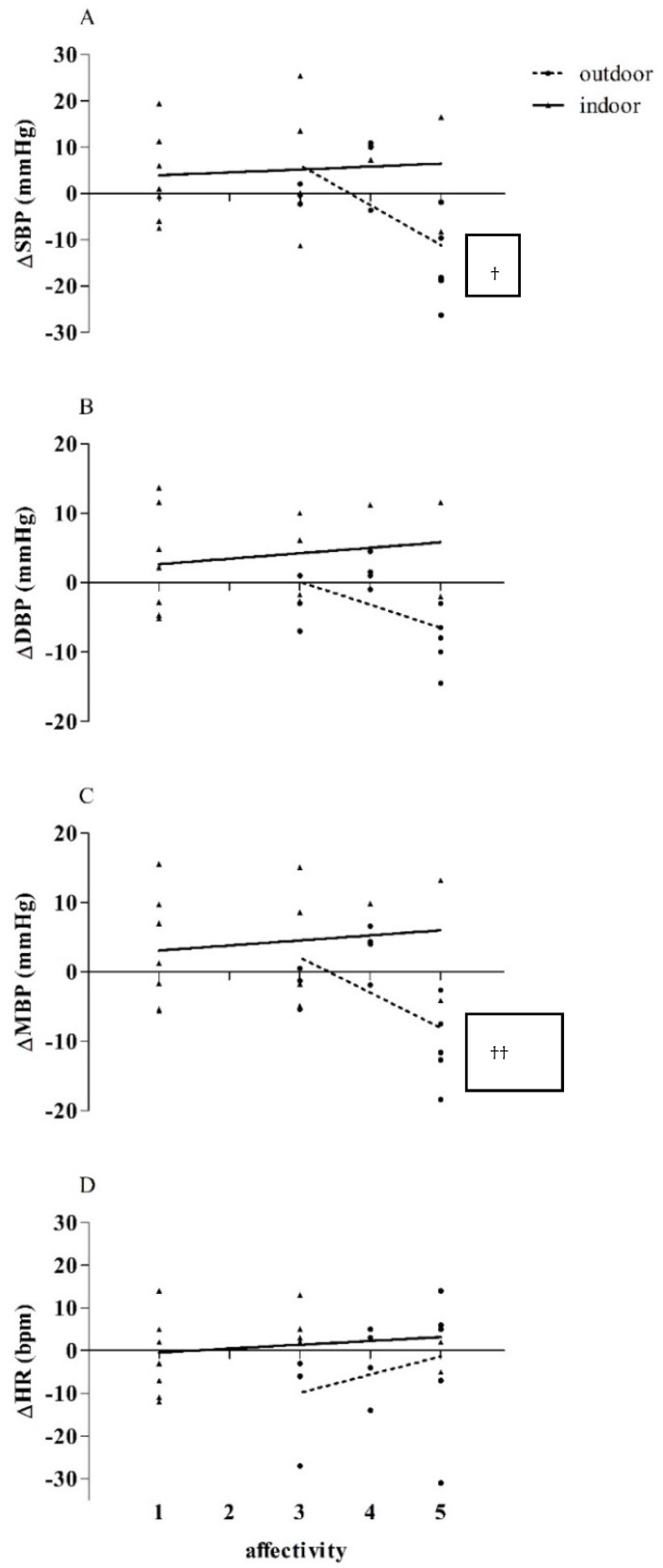
Correlation between the effect of training on BP, HR, and affective response after nine weeks of combined indoor and outdoor training. ∆ = difference between the post 9 weeks value and the baseline value. ^†^—indicates values of r = −0.60 and *p* = 0.03 for SBP of the OCT group. ^††^—indicates values of r = −0.62 and *p* = 0.02 for MBP of the OCT group.

**Table 1 ijerph-19-16009-t001:** General characteristics of the sample (*n* = 26).

Variables	OCT (*n* = 12)	ICT (*n* = 14)	*p*-Value
	$\bar{x}$ ± sd	$\bar{x}$ ± sd	
Age (years)	67.3 ± 8.2	65.8 ± 7.8	0.644
Body mass (kg)	78.1 ± 15.2	79.3 ± 13.9	0.839
Height (m)	1.68 ± 0.1	1.64 ± 0.1	0.390
BMI (kg/m²)	27.4 ± 3.0	29.4 ± 4.3	0.198
	*n* (%)	*n* (%)	
Sex (female)	3 (25.0)	7 (50.0)	0.192
Hypertension	6 (50.0)	11 (78.6)	0.126
Dyslipidemia	8 (66.7)	5 (35.5)	0.115
Diabetes	3 (25.0)	2 (14.3)	0.489
Antihypertensive users			
No medication	7 (58.3%)	5 (35.7)	0.248
Up to 1 medication	3 (25.0)	5 (35.7)	0.555
Up to 2 medications	1 (8.3)	4 (28.6)	0.191
Up to 3 medications	1 (8.3)	0 (0.0)	0.270

Note: OT—outdoor training group. IT—indoor training group. $\bar{x}$
± sd—mean ± standard deviation. BMI—body mass index.

**Table 2 ijerph-19-16009-t002:** Effects of combined Outdoor and Indoor training on BP and HR (*n* = 26).

	OCT (*n* = 12)			ICT (*n* = 14)			*p*-Value		
	Pre	Post	Cohen’s d	Pre	Post	Cohen’s d	g	t	g*t
	$\bar{x}\;$ ± sd	$\bar{x}$ ± sd		$\bar{x}\;$ ± sd	$\bar{x}$ ± sd				
SBP (mmHg)	122.7 ± 9.1	118.8 ± 10.3	0.40	123.4 ± 14.0	128.2 ± 19.9	−0.27	0.341	0.885	0.312
Δ SBP	−3.9 ± 11.7			4.8 ± 11.1					
DBP (mmHg)	69.7 ± 5.9	65.9 ± 6.9 ^†^	0.59	71.3 ± 6.2	75.0 ± 6.5	−0.58	0.004	0.876	0.040
Δ DBP	−3.8 ± 5.3			3.7 ± 6.9					
MBP (mmHg)	87.4 ± 6.0	83.5 ± 6.8	0.61	88.6 ± 6.9	92.7 ± 9.6	−0.49	0.015	0.836	0.065
Δ MBP	−3.9 ± 7.3			4.1 ± 8.0					
HR (bpm)	70.8 ± 12.3	65.9 ± 6.9	0.49	73.1 ± 8.2	73.9 ± 9.6	−0.09	0.057	0.481	0.284
Δ HR	−4.9 ± 12.7			0.8 ± 7.8					

SBP—systolic blood pressure. DBP—diastolic blood pressure. MBP—mean blood pressure. HR—heart rate. $\bar{x}$
± sd—mean ± standard deviation. g—group. t—time. g*t—interaction group vs. time. ^†^—*p* ≤ 0.05 vs. indoor group at post-training. Delta (∆)—changes from pre to post-intervention.

**Table 3 ijerph-19-16009-t003:** Rating Perception Effort of the outdoor and indoor training groups before and after nine weeks of training (*n* = 26).

	OCT (*n* = 12)			ICT (*n* = 14)			*p*-Value		
	Pre	Post	Cohen’s d	Pre	Post	Cohen’s d	g	t	g*t
	$\bar{x}\;$ ± sd	$\bar{x}$ ± sd		$\bar{x}\;$ ± sd	$\bar{x}$ ± sd				
RPE	4.08 ± 1.44	3.75 ± 1.05	0.26	3.75 ± 1.05	3.35 ± 1.59	0.30	0.144	0.202	0.671

Notes: OT—outdoor training. IT—indoor training. RPE—Rating of Perception Effort. $\bar{x}$
± sd: mean ± standard deviation. g—group. t—time. g*t—interaction group vs. time.

## Data Availability

Data are available at any time by contacting the authors.

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
