# Peer review of "Combined Training with Aerobic Exercise Performed Outdoors Can Promote Better Blood Pressure and Affective Responses in Individuals with Cardiovascular Risk Factors"

_ijerph, 2022, doi:10.3390/ijerph192316009_

Round 1

Reviewer 1 Report

Abstract

If the wordcount allows it, I would encourage the authors to indicate number of participants that took part in the study in the ‘methods’ section of the abstract, and also add a bit more detail, if possible (e.g. their age (and not just adult or elderly), level of fitness?).

Introduction

Appreciate authors’ focus is on Brazil and its population, but perhaps some stats on prevalence of hypertension in the world may help engage a wider audience.  

L26 PNAD is not a common abbreviation recognised worldwide, and also is not used again. Consider removing?

L56 Change from Rate to Rating (of perceived exertion)

L46 – 54. I am not sure there is a clear rationale as to why we may expect to see a difference in the response in BP after training indoors vs outdoors?

L64. I am not sure this sentence works. Are you suggesting that environment (outdoor vs indoor) is important (if so, why the need for this study)?

Methods

Section 2.2. Please state the inclusion/exclusion criteria.

I’d personally report the characteristics of the participants in methods (and not results), but that may be a matter of opinion. What I am not too sure about is the adult vs elderly – did you treat them as one group? Also, were these participants healthy / at risk of hypertension (so your focus is on prevention), or have they already been diagnosed with hypertension (so focus is more on treating the disease)

Section 2.4. I understand randomisation may not have been possible. Did you take any steps to ensure (or account for) this fact in your results or data analysis? E.g. did you take baseline levels into consideration? As you say participants were already performing physical training, how confident are the authors that a similar background / training history was present in both groups?

L94 – ‘in the second moment’ does not seem to be a common English expression?

The intervention section 2.5.5 may need some further detail. For example, you state: The participants already performed physical training in the PROCOR extension project. For how long had participants been training when the intervention started? Was that similar in both groups?

What was the exercise in AT? E.g. in the ICT you suggest both? So was this at random, by participants preference… I also would like to ask whether you can confidently compare these two training programmes, as the differences seem to extend beyond the environment.

What was the intensity quantified / defined? In other words, how was ‘high’ intensity defined?

I am a bit confused as to whether the exercise intervention in both groups were identical? The only difference was where the exercise session took place? It reads as if it the intensity was ‘self-selected’ – so that participants selected an intensity that matched the RPE?

Results

You keep referring to elderly (e.g. 84.6% of your sample was ‘elderly’). How do you define old?

Discussion

You argue that the intensity was the same in both OCT and ICT based on the fact that it was prescribed using RPE. We know RPE is useful, but has its limitations, so wondering if you can address this?

For example, in line 285 you indicate how exercising outdoors results in a reduced RPE (compared to the same absolute intensity but indoors, I assume). Therefore if we can clarify how exercise intensity has been prescribed (ie if using RPE to prescribe intensity of exercise?), would it not be possible that exercising outdoors resulted in an increased absolute intensity, thus explaining the enhanced responses in the OCT group?

You seem to focus on the psychological aspects of exercising outdoors, such how affection can enhance positive effects of training. I wonder if you could also consider the possibility that the outdoor training programme has some additional physiological benefits (e.g. related to sun exposure and its effect on Vit D and then BP).

Line 304 – this sentence does not make sense to me: The present study has as strong points the training period

Author Response

Thank you very much for your comments, please see the attachment.

Reviewer 2 Report

It was my pleasure to review the article entitled Combined Training with Aerobic Exercise Performed Outdoors Promotes Better Blood Pressure and Affective Responses in Individuals with Cardiovascular Risk Factors. The study is a non-randomized comparative clinical trial from PROCOR program included 26 participants. The results shown that combined training outdoor seems to be more effective to decrease diastolic blood pressure and also are associated with better affective responses. The article is well written and easy to read. In the last paragraph of the discussion, I suggest to add that the small sample could be a limitation.

Author Response

(The authors gave the same response as above.)

Reviewer 3 Report

General comments

I thank the authors for the opportunity to review their work. The manuscript describes an exercise training study where the effects of combined aerobic training that was conducted in an outdoor setting was compared with combined aerobic exercise conducted in an indoor setting on cardiovascular risk factors (BP, heart rate) and affective response. 26 participants who were mostly elderly and male completed 3 x per week exercise sessions for a period of 9 weeks. Results indicated that following outdoor combined training, participants had significantly lower diastolic BP. Results also indicated an inverse correlation between affective response and systolic and mean BP, with outdoor aerobic training. The manuscript is reasonably well written though I would suggest a revision of a few phrases throughout. The major concern I have with the study is the lack of details surrounding certain elements of the design. The study seems to be described slightly differently in different sections of the paper. Of primary concern is the allocation to treatment groups. If there was no randomization and participants seem already to be trained given their involvement in the PROCOR program, why not consider a cross over design where participants engage in both ICT and OCT for 9 weeks before changing over. Overall some of the language should be toned down regarding the significance of the findings given the concerns about threats to validity. There are some interesting concepts that warrant further examination.

Specific comments

Please see specific concerns and suggestions below.

Abstract

1.     While it is mentioned in the introduction, it is not clear in the abstract that combined training refers to aerobic and resistance training. Please include a brief description in the abstract as well. 

2.     While the findings are interesting, it would seem that with only 26 participants that the authors could be more cautious in their interpretations?

Introduction

3.     In paragraph 1  perhaps instead of “the hypertension” you could simply state “hypertension”.

4.     Page 2, paragraph 3, “rate perception effort” can be changed to “rating of perceived exertion”.

5.     While it is assumed from the description in the introduction, could the authors clarify that the resistance training portion of the intervention was conducted indoors? If so, were any measures of affect and or RPE taken during or following the resistance exercise portion or only at the end of the entire exercise session? It seems aerobic exercise was done first (indoors or outdoors) then all participants moved indoors to complete resistance exercise after which there was a RPE and affective measurement taken? 

6.      

Methods

7.     It is described that the study is non-randomized. Can this be stated in the abstract?

8.     In section 2.5.2 can the authors clarify that the BP measures used for outcomes assessment were taken on a Monday, Wednesday and Friday, before and after the planned exercise sessions? Were all measures taken indoors or outdoors for the OCT group? 

9.     In sections 2.5.3 RPE is described as a 1-10 scale and then on page 4 paragraph 4 it is stated the 6-20 scale is used. This is confusing and I am not clear whether the 6-20 scale was used to determine training intensity during each session depending on whether the target was moderate vs high intensity. Was the 1-10 scale then used for the measures following an entire session? Please clarify

10.  Can the authors clarify what happened if there was rain or poor weather during a planned OCT session? Can the authors state whether adjustments were needed at any time over the 9 weeks.

11.  Can the authors describe how participants were approached about doing their sessions either indoors or outdoors? It seems the 2 sessions were done close together. Did ICT or OCT participants see members from the other group and ever ask to be switched or do do the other mode of training?

Results

12.  In table 2, the results seem to indicate that mean BP in the indoors group got worse over the period of the intervention. Why would this be? 

13.  On page 6 lines 222-224 the statistically significant difference in affective response to training. Is this between the groups (OCT vs ICT) following the 9 weeks of the program?

Discussion

14.  The language in the first paragraph of the discussion is fairly strong, considering the non-randomized sample and small numbers of participants. It would be preferrable to see these remarks tempered given the limitations in the study design.

15.  The results with affect are interesting. Can the authors elaborate on why they think DBP was improved in the OCT group but affect was only related to SBP and MBP not DBP especially since they discuss vascular stress.

16.  The aspects of OCT that may seem to provide a benefit in terms of affect are described as exposure to nature. Can the authors describe the environment surrounding the outdoor track where activity took place? Were there other people exercising at that time, urban city like environment or more natural with trees etc. ?

17.  In the final paragraph of the discussion the authors mention that the groups self-selected based on whether they preferred to exercise indoors or outdoors. This a major limitation of the study ass earlier it was discussed that allocation was based on the time of day the sessions took place and availability of light etc. If self-selection took place this should be mentioned in the allocation section.

18.  The authors also mention 9 weeks as a strength of the study. Why not 12 weeks? Or 6 months, particularly if these participants are described as already trained (at least 3 months)?

Tables and Figures

NA

Author Response

(The authors gave the same response as above.)

Reviewer 4 Report

Dear Editor and Authors,
I read the article entitle "Combined Training With Aerobic Exercise Performed Outdoors Promotes Better Blood Pressure And Affective Responses In Individuals With Cardiovascular Risk Factors".
This is a very well-written, high-quality manuscript. This research has clinical relevance and highlights sensitive healthcare components that need to be addressed in order to improve the cardiovascular risk factors. Results were clearly and succinctly presented. This article should be considered for publication. Only two observations:
- Figure 2 is difficult to understand, perhaps a summary table of the pressure and frequency deltas in the two groups analyzed would be better; this table would allow a better evaluation of the efficacy of the two types of exercise on blood pressure and heart rate;
- the authors should indicate in the study limitations that about half of normotensive patients were included in the study (12 of 26 patients were not taking any antihypertensive therapy); in fact, the indicated pressure variations are in any case within the normal limits. Probably it would have been more correct to consider only hypertensive patients but the small quantities of the sample would have skewed the statistics.
Best regards

Author Response

(The authors gave the same response as above.)

Reviewer 5 Report

This study investigated the blood pressure and basal heart rate of individuals (old adults and elderly) with cardiovascular risk after a combined aerobic and resistance training program with aerobic exercise performed indoor (ICT) or outdoor (OCT). The conclusion seems to be overestimated by the authors. Besides an interesting association between the delta of blood pressure and affectivity to training, the affirmation that the combined training with outdoor aerobic exercise is better for reducing blood pressure is debatable. There is not intra-group change on blood pressure after training and the effect size is small. It may be premature to reach such conclusion based only on the inter-group difference due to the sample heterogeneity of genre, age and cardiovascular risk factors.

Introduction

This section was well written; however, some aspects can be improved. The introduction can be more informative regarding the possible effects of distinct models of training (aerobic, resistance and combined) on blood pressure. Additionally, a short hypothesis for the possible outcomes of the present study should be introduced based on the literature revision.

Methods

The study was well conducted and the authors adopted good practices for intensity prescription and load monitoring by using the subjective scale of rate perception effort to individualize training.

An exclusion criteria should be included based on training adherence (e.g. minimum adherence value).

Was the sample power, normality and homoscedasticity of the data analyzed? Why using a parametric test for the analysis of variance (ANOVA two-way) and a non-parametric test (Spearman’s) for verifying the association among variables?

Results

For most part, the results are clearly presented.

The effect size is not presented on table 3.

Did the authors verified the association between blood pressure and affectivity response on the overall sample? Maybe the significant association could still be observed without the group allocation.

Discussion and conclusions

In my opinion the title, discussion and conclusion should be rewritten to shed more light on the association between the delta of blood pressure and affective response to training, while the inter-group difference of blood pressure found after training should be carefully interpreted since there is not within-group effect of training.

A discussion between expected (hypothesis based on the literature) and found results would be welcome here.

The authors raised some valid limitations of the study, but one study limitation not mentioned is the sample heterogeneity due to the lack of inclusion criteria for genre, age or cardiovascular risk factor and medication. Moreover, there is not exclusion criteria for training adherence.

I must disagree with the statement of the last paragraph and conclusion since nine weeks of training did not promote significant changes on blood pressure of both groups.

Author Response

(The authors gave the same response as above.)

Round 2

Reviewer 5 Report

I thank the authors for the cover letter and I appreciate your comments. The authors have done a good job addressing most of my previous questions and comments. All changes throughout the document makes this an improved version of the manuscript. However, there remain some areas where I would like to suggest improvement.

1.     In the title, change “can promotes” to “can promote”

2.     Throughout the text, change “non-randomized way” to “non-randomized design”

3.     I would like to request the inclusion of the following information:

a.      The information about the analysis of data normality in the Data Analysis session

b.     The standard deviation for the training adherence as result

c.      After the first paragraph of the discussion, a brief and possible justification for the hypothesis of BP reduction have not been confirmed for the ICT

d.     I suggest to insert in the conclusion a sentence stating that the inter-group difference of blood pressure found after training should be carefully interpreted since there is not within-group effect of training.

Author Response

We appreciate your comments, please check our attached answers. Thank you so much.
